# Foot-and-Mouth Disease Virus Capsid Protein VP1 Antagonizes Type I Interferon Signaling via Degradation of Histone Deacetylase 5

**DOI:** 10.3390/cells13060539

**Published:** 2024-03-19

**Authors:** Qing Gong, Shanhui Ren, Yongxi Dou, Berihun Afera Tadele, Tao Hu, Luoyi Zhou, Tao Wang, Kaishen Yao, Jian Xu, Xiangping Yin, Yuefeng Sun

**Affiliations:** 1State Key Laboratory for Animal Disease Control and Prevention, College of Veterinary Medicine, Lanzhou University, Lanzhou Veterinary Research Institute, Chinese Academy of Agricultural Sciences, Lanzhou 730000, China; 15682780368@163.com (Q.G.); renshanhui@caas.cn (S.R.); douyongxi@caas.cn (Y.D.); berihun.afera@mu.edu.et (B.A.T.); zhouluoyi789@163.com (L.Z.); taowang2016@lzu.edu.cn (T.W.); yaokaishen0802@163.com (K.Y.); xujian@caas.cn (J.X.); 2The First Clinical Medical College, Gansu University of Chinese Medicine, Lanzhou 730000, China; kmdhutao@sina.com; 3College of Animal Science and Technology, Hebei Normal University of Science & Technology, Qinhuangdao 066600, China; 4School of Dentistry, Lanzhou University, Lanzhou 730000, China

**Keywords:** innate immune response, FMDV infection, VP1, HDAC5, IFN-β

## Abstract

Foot-and-mouth disease (FMD) is a highly contagious and economically important disease of cloven-hoofed animals that hampers trade and production. To ensure effective infection, the foot-and-mouth disease virus (FMDV) evades host antiviral pathways in different ways. Although the effect of histone deacetylase 5 (HDAC5) on the innate immune response has previously been documented, the precise molecular mechanism underlying HDAC5-mediated FMDV infection is not yet clearly understood. In this study, we found that silencing or knockout of HDAC5 promoted FMDV replication, whereas HDAC5 overexpression significantly inhibited FMDV propagation. IFN-β and IFN-stimulated response element (ISRE) activity was strongly activated through the overexpression of HDAC5. The silencing and knockout of HDAC5 led to an increase in viral replication, which was evident by decreased IFN-β, ISG15, and ISG56 production, as well as a noticeable reduction in IRF3 phosphorylation. Moreover, the results showed that the FMDV capsid protein VP1 targets HDAC5 and facilitates its degradation via the proteasomal pathway. In conclusion, this study highlights that HDAC5 acts as a positive modulator of IFN-β production during viral infection, while FMDV capsid protein VP1 antagonizes the HDAC5-mediated antiviral immune response by degrading HDAC5 to facilitate viral replication.

## 1. Introduction

Foot-and-mouth disease (FMD) is the most serious disease affecting livestock, resulting in substantial economic losses for animal husbandry [1,2,3,4]. There are seven serotypes of FMD virus (FMDV) circulating worldwide namely O, A, C, Asia, SAT1, SAT2, and SAT3, as well as numerous other subtypes. The disease can be prevented through vaccination of livestock but the development of vaccines against FMDV faces numerous challenges, primarily due to the limited or non-existent cross-protection across serotypes and subtypes [5,6,7]. The FMDV genome has a length of approximately 8.5 kb and contains a genetic code for four structural proteins: VP1, VP2, VP3, and VP4. Additionally, it includes the genetic information for eight non-structural proteins: L, 2A, 2B, 2C, 3A, 3B, 3C, and 3D [8,9,10]. Among these viral proteins, VP1, VP2, and VP3 make up the entire outer capsid surface, whereas VP4 contributes to the structural integrity of the inner capsid [2]. Studies have shown that the FMDV VP1 protein is a crucial component of the capsid, which is involved in the attachment and entrance of the virus to host cells via an arginine-glycine-aspartic acid (RGD) sequence in the G-H loop. The FMDV VP1 protein suppresses the type I interferon response by interacting with sorcin and antagonizes the TPL2-mediated IRF3/IFN-β signaling pathway [11,12]. The FMDV VP1 83E site is involved in the interaction with MAVS, and this specific interaction inhibits interferon responses by destroying the interaction between MAVS and TRAF3 [13]. Moreover, DNAJ3 inhibits FMDV replication by interacting with and degrading VP1 via the lysosomal degradation pathway, as well as attenuating its antagonistic role in the IFN-β signaling pathway [14].

Post-translational modifications, such as ubiquitination, SUMOylation, and phosphorylation, were found to be involved in FMDV infection. FMDV infection was inhibited by the suppression of SUMOylation on the 3C function and ubiquitination of TRIM21, but the propagation was promoted by FMDV separately degrading Vps28, RIG, and MDA5 in different ways, and inhibiting the phosphorylation of IRF3 [15,16,17,18,19]. Histone deacetylation, one of the post-translational modifications, has been reported to be significant in viral infection [20,21]. Histone acetylation regulates the immune response induced by the bone marrow-derived mast cell recognition of FMDV virus-like particles [22]. ID1 inhibits the transcriptional activity of FOXO1 through HDAC4-mediated deacetylation, thus promoting IFN-I production and antiviral immune responses. FMDV antagonizes the function of ID1 by facilitating its degradation via Cdh1-mediated ubiquitination to promote viral reproduction [23]. In a previous study, FMDV VP3 was found to interact with HDAC8 and to promote autophagic degradation to facilitate viral replication [24]. However, whether the vital FMDV capsid protein VP1 affects the immune response by regulating histone acetylation during viral infection remains unclear.

In this study, we found that the replication of FMDV was increased in HDAC5-silencing or knockout cells, whereas it was decreased in HDAC5-overexpressed cells. We also found that HDAC5, a positive regulator of the expression of IFN-β and ISGs, promoted the activation of the IRF3/IFN-β signaling pathway, while FMDV VP1 degraded HDAC5 through the proteasome pathway to facilitate viral replication. Our study provides new insights into the therapeutic potential for developing new antiviral strategies.

## 2. Materials and Methods

### 2.1. Cells, Virus, and Infection

Human embryonic kidney 293T (HEK293T), porcine kidney (PK-15), and baby hamster kidney-21 (BHK-21) cells were obtained from ATCC (GNHa10, BH-C706, and SCSP-502, respectively) and cultured in Dulbecco’s modified DMEM medium (Solarbio, 11995, Beijing, China) supplemented with 10% fetal bovine serum (FBS) (Gibco, 10099141C, Waltham, MA, USA) and 1% penicillin-streptomycin (Beyotime, C0222, Shanghai, China). An HDAC5-KO BHK-21 cell line was generated using CRISPR/Cas9 technology [25]. To generate the HDAC5-KO cell line, double-stranded oligonucleotides corresponding to the target sequences were cloned into the pSpCas9 (BB) plasmid (sgRNA CCCGTAGCGCAGGGTCCATG). The plasmids with the correct sequence were transfected into BHK-21 cells and incubated for 48 h. Cells were further cultured in DMEM (10% FBS) supplemented with 1 μg/mL puromycin (Beyotime, ST551, Shanghai, China) and the cell medium was replaced every 48 h. Single colonies were selected to establish clonal cell lines after puromycin screening. Western blotting and sequencing were performed to identify monoclonal cell lines.

The FMDV O/BY/2010 strain and Sendai virus (SeV) were maintained at the National Foot-and-Mouth Disease Reference Laboratory of the Lanzhou Veterinary Research Institute. The VSV and VSV-GFP viruses were stored in our laboratory. Cells were infected with viruses (FMDV, VSV, and VSV-GFP, MOI = 0.05; SeV,50 HAU/mL) and incubated for 1 h. Subsequently, the cells were cultured in a fresh medium without FBS. The supernatants were then harvested at the indicated time points, and the titers of viruses were quantified using TCID_50_.

### 2.2. Reagents, Antibodies, and Plasmids

Polyinosinic-polycytidylic acid (poly(I:C)) was purchased from InvivoGen (tlrl-picw, Hongkong, China). HDAC5 and control siRNAs were obtained from GenePharma (A100001, Shanghai, China). Primary and secondary antibodies, including anti-phosphorylated IRF3 (CST, #4947, Danvers, MA, USA), anti-total IRF3 (CST, #4302, Danvers, MA, USA), anti-β-actin (Sigma-Aldrich, A5316, St. Louis, MO, USA), anti-GFP (TRANS, HT801-01, Beijing, China), anti-Flag (Abmart, M20008, Shanghai, China), anti-Myc (Santa Cruz Biotechnology, sc-40, Dallas, TX, USA), anti-mouse IgG (H + L), F(ab′)2 Fragment (Alexa Fluor^®^ 488 Conjugate, CST, #4408, Danvers, MA, USA), and anti-rabbit IgG (H + L), F(ab′)2 Fragment (Alexa Fluor^®^ 594 Conjugate, CST, #8889, Danvers, MA, USA) were purchased from the indicated manufacturers. The antibody against the FMDV VP1 protein was provided by Professor Haixue Zheng. MG132 (20 μM) (Merk & Co., M8699, Darmstadt, Germany), chloroquine diphosphate (CQ) (100 μM) (Sigma-Aldrich, C-271, St. Louis, MO, USA), NH_4_Cl (5 M) (Sigma-Aldrich, 9434, St. Louis, MO, USA), 3-methyladenine (3-MA) (10 μM) (Sigma-Aldrich, 189490, St. Louis, MO, USA), DMSO (Sigma-Aldrich, D8414, St. Louis, MO, USA), and DAPI (Sigma-Aldrich, 10236276001, St. Louis, MO, USA) were purchased from the indicated manufacturers. pcDNA3.1, HDAC5-Myc, VP0-Flag, VP1-Flag, VP3-Flag, 2A + 2B-Flag, 2B-Flag, 2C-Flag, 3A-Flag, 3C-Flag, and 3D-Flag were constructed and stored in our laboratory.

### 2.3. Coimmunoprecipitation and Western Blotting

Coimmunoprecipitation was performed to confirm the interaction between HDAC5 and VP1 in HEK293T cells. HEK293T cells (6 × 10^6^) were seeded on 100 mm dishes overnight. Cells were transfected separately with 5 μg HDAC5-Myc, 5 μg VP1-Flag, and 5 μg empty-vector as indicated, and incubated for 36 h. Subsequently, the cells were harvested separately and lysed in NP40 buffer (Beyotime, P0013F, Shanghai, China) supplemented with 1% protease inhibitor. The supernatants were collected after centrifugation at 13,000× *g* rpm for 20 min and incubated with anti-Myc or anti-Flag overnight at 4 °C. The lysate–antibody complexes were incubated with protein G beads (cytiva, 17061801, Cardiff, UK) for 4–6 h at 4 °C. The beads were wash with cold NP40 buffer for five times and eluted with sodium dodecyl sulfate (SDS) loading buffer (Solarbio, P1016, Beijing, China) by boiling for 10 min. Protein samples were separated by sodium dodecyl sulfate-polyacrylamide gel electrophoresis (SDS-PAGE), and analyzed by Western blotting.

For Western blotting, cells were lysed in RIPA buffer (Beyotime, P0013B, Shanghai, China) supplemented with 1% protease inhibitor (Beyotime, P1006, Shanghai, China). The lysates were centrifuged at 13,000× *g* rpm for 20 min to remove cellular debris and nuclei. Protein samples were obtained by denaturing the supernatants with sodium dodecyl sulfate (SDS) loading buffer (Solarbio, P1016, Beijing, China) at a high temperature for 10 min. Next, the samples were separated using SDS-PAGE and subsequently transferred to polyvinylidene fluoride membranes. To prevent non-specific interactions, 5% skim milk was used for blocking for 60 min at room temperature. The primary antibodies were incubated overnight as follows: anti-HDAC5 (1:1000), anti-β-actin (1:4000), anti-VP1 (1:2500), anti-p-IRF3 (1:1000), anti-IRF3 (1:1000), anti-Myc (1:2500), anti-Flag (1:3000), anti-GFP (1:3000). The secondary antibodies were incubated for 60 min. Lastly, antibody–antigen interactions were detected using a chemiluminescence reagent (Beyotime, P0203, Shanghai, China).

### 2.4. Reporter Gene Assays

HEK293T cells (1 × 10^5^) were seeded on a 48-well culture plate overnight. Firefly luciferase reporter (pIFN-β-luc, pISRE-luc, 100 ng), pRL-TK (10 ng), and other indicated plasmids (100 ng) were co-transfected into HEK293T cells and incubated for 24 h. Cells were detected without virus infection, or after SeV infection for 12 h. Dual-luciferase assays were conducted to detect the activity of the IFN-β or IFN-stimulated response element (ISRE) reporters.

### 2.5. RNA Extraction and RT-qPCR

mRNA expression was examined using real-time qPCR. Total RNA was extracted using the TRIzol reagent (Sangon Biotech, B610409, Shanghai, China). An SYBR Green RT-qPCR kit (Takara, RR037, Kyoto, Japan) was used to quantify specific mRNAs. Each sample’s β-actin expression was used to normalize the data. All experiments were conducted in triplicate. The 2^−ΔΔCt^ method was employed to analyze relative expression changes [26]. The primer sequences used in this study are listed in Table 1.

### 2.6. Cell Treatments, Transfection, and ELISA

PK-15, BHK-21, and HEK293T cells were transfected with plasmids, siRNA, or poly(I:C) (2 μg/mL) using JetPRIME Polyplus reagent (Polyplus Transfection, 101000046, Strasbourg, France).

PK-15 cells were transfected separately with siCtrl and siHDAC5, and incubated for 36 h. The transfected cells were infected with FMDV. Culture supernatants were harvested at the indicated time points and detected by ELISA. WT and HDAC5-KO BHK-21 cells were infected separately with FMDV. Culture supernatants of WT and HDAC5-KO BHK-21 cells were harvested at the indicated time points and detected by ELISA. HEK293T cells were transfected separately with empty-vector and HDAC5 (2 μg), and incubated for 36 h. The transfected cells were infected with VSV. Culture supernatants were harvested at the indicated time points and detected by ELISA. The production of IFN-β in PK-15, HEK293T, and BHK-21 cells was detected using ELISA according to the manufacturer’s instructions (Elabscience, E-EL-H0085c, Wuhan, China; JYM, JYM0026Hu, Wuhan, China; Cloud-Clone, SEA222Mu, Wuhan, China; respectively).

### 2.7. RNA Interference

siHDAC5 and siCtrl sequences were designed by Suzhou GenePharma Co., Ltd (Shanghai, China). The siRNA sequences are listed in Table 2.

### 2.8. Statistical Analysis

The experiments were repeated thrice to ensure data accuracy. All analyses, including unpaired two-tailed Student’s *t*-tests, were performed using GraphPad Prism software (version 6.0, GraphPad Software, Boston, MA, USA). Significantly different means are indicated by asterisks. Statistical significance is expressed as follows: No significant (ns), * *p* < 0.05, ** *p* < 0.01, *** *p* < 0.001, and **** *p* < 0.0001.

## 3. Results

### 3.1. HDAC5 Inhibits FMDV Replication

To study the impact of HDAC5 on FMDV infection, PK-15 cells were transfected with three siRNAs specifically targeting HDAC5 or a control siRNA (siHDAC5 vs. siCtrl). Subsequently, qPCR and Western blotting were used to detect HDAC5 mRNA and protein levels, respectively. Among the three siRNAs, siRNA3 (siHDAC5) showed the most efficient silencing of HDAC5 mRNA, and was, therefore, used in this study to knockdown the expression of HDAC5 (Figure 1A,B).

To investigate the potential role of HDAC5 in FMDV replication, siHDAC5 or siCtrl was transfected into PK-15 cells, followed by viral infection. FMDV VP1 is an essential protein for FMDV replication and is widely used as a marker for FMDV production [11]. In this study, we found that compared with controls, the expression and transcription of FMDV VP1 were markedly increased in HDAC5-silenced PK-15 cells (Figure 1C,D), which was further supported by the viral titers of FMDV in HDAC5-silenced PK-15 cells (Figure 1E).

We further assessed the propagation of FMDV in HDAC5-KO BHK-21 cells. An HDAC5 knockout (KO) cell line was generated in BHK-21 using CRISPR/Cas9. To generate the HDAC5-KO cell line, double-stranded oligonucleotides corresponding to the target sequences were cloned into the pSpCas9 (BB) plasmid, which was transfected into BHK-21 cells. Single colonies were selected to establish clonal cell lines after puromycin screening. DNA sequencing results showed that thirteen bases of the first exon of one allele of HDAC5 were deleted in the HDAC5 knockout (KO) cell line (Appendix A). The expression of HDAC5 was completely blocked in the KO cells (Appendix A), and there was no significant difference in viability between WT and HDAC5-KO cells (Appendix A).

As shown in Figure 1F–H, both the expression of FMDV VP1 and mRNA were increased in HDAC5-KO BHK-21 cells, as well as the viral titers of FMDV. VSV is generally considered to be a model virus for RNA virus research. Moreover, to further investigate whether HDAC5 regulates the replication of other viruses, we also examined VSV replication in HDAC5-KO BHK-21 cells. As shown in Appendix A, the expression of VSV labelling GFP protein and mRNA of structural G protein were significantly increased in HDAC5-KO BHK-21 cells compared with control groups, as well as the titers of VSV. These results indicate that HDAC5 inhibits the replication of RNA viruses, such as FMDV and VSV, during viral infection.

### 3.2. HDAC5 Enhances IFN-β and ISGs Expression during FMDV Infection

Previous studies have shown that HDACs are involved in regulating the IFN-I signaling pathway during viral infection [23,27,28]. To further investigate the role of HDAC5 in regulating the IFN-I signaling pathway, we examined whether IFN-β expression and production are affected by HDAC5 in HEK293T cells. As shown in Figure 2A,B, after SeV stimulation, the overexpression of HDAC5 enhanced the activity of IFN-β and ISRE luciferase reporter. Next, we examined the function of HDAC5 knockdown on the production of IFN-β and ISGs upon FMDV infection in PK-15 cells. The data indicated that HDAC5 knockdown significantly inhibited the mRNA expression levels of IFN-β and ISGs and increased the viral genomic copy numbers of FMDV (Figure 2C–E,I). Furthermore, the production of IFN-β in the HDAC5 knockdown cell was greatly decreased compared with the control group upon FMDV stimulation (Figure 2J). Since poly(I:C) is a synthetic analog of double-stranded RNA, we also studied the effect of HDAC5 knockdown on the production of IFN-β and ISGs after poly(I:C) transfection in PK-15 cells. As shown in Figure 2F–H, we found that the knockdown of HDAC5 also markedly inhibited the mRNA expression levels of IFN-β and ISGs.

To determine whether HDAC5 regulates the production of IFN-β and ISGs in HDAC5-KO BHK-21 cells upon stimulation by FMDV and poly(I:C), we analyzed the mRNA expression levels of IFN-β and ISGs. As shown in Figure 3A–F, the knockout of HDAC5 significantly suppressed the mRNA levels of IFN-β, ISG15, and ISG56 in BHK-21 cells compared with the control groups, as well as the secretion of IFN-β protein (Figure 3I, and markedly promoted the viral genomic copy numbers and viral titers of FMDV (Figure 3G,H). To explore whether HDAC5 regulates the production of IFN-β and ISGs after VSV stimulation, we repeated the above experiments in HDAC5-KO BHK-21 cells infected with VSV instead of FMDV. The knockout of HDAC5 resulted in the inhibition of IFN-β, ISG15, and ISG56 transcription levels, and increased the viral genomic copy numbers of VSV (Appendix A). At the same time, the secretion of IFN-β was highly decreased compared with the control groups (Appendix A). Next, we investigated the synthesis of IFN-β and ISGs in HEK293T cells transfected with HDAC5 plasmid and treated with poly (I:C) or VSV. The overexpression of HDAC5 significantly promoted the synthesis of IFN-β, ISG15, and ISG56 stimulated by poly (I:C) or VSV (Appendix A) and inhibited the viral genomic copy numbers of VSV (Appendix A). The production of IFN-β was notably increased in HDAC5-overexpressed cells (Appendix A). Taken together, these results show that HDAC5 activates the IFN-β signaling pathway during FMDV and VSV infection.

### 3.3. HDAC5 Regulates FMDV-Induced Phosphorylation of IRF3

To better understand the potential mechanism of HDAC5 in antiviral innate immunity, we investigated the effects of HDAC5 on the activation of the IFN-β promoter caused by specified elements, including RIG-I, MDA5, MAVS, TBK1, IKKε, and IRF3. We found that HDAC5 promoted the transcriptional activity of IFN-β stimulated by these plasmids, respectively (Figure 4A–F). IRF3 is essential for IFN-β expression and the antiviral response [29,30,31]. Compared with the control groups, the overexpression of HDAC5 remarkably promoted the phosphorylation of IRF3 upon FMDV infection (Figure 4G). The deficiency of HDAC5 strongly inhibited the phosphorylation of IRF3 in PK-15 cells upon FMDV infection, as well as in HDAC5 knockout BHK-21 cells (Figure 4H,I). Meanwhile, we detected phosphorylation of IRF3 in cells infected with VSV; Knockout of HDAC5 significantly reduced the phosphorylation of IRF3 (Appendix A). These results indicate that HDAC5 promotes IRF3-mediated antiviral innate immune responses during both FMDV and VSV infection.

### 3.4. FMDV Capsid Protein VP1 Targets and Degrades HDAC5 through the Proteasome Pathway

Studies have reported that the proteins of FMDV hinder the IFN-β signaling pathway [14,32,33]. To further investigate which viral protein regulates HDAC5 and counteracts the IFN-β-mediated signaling pathway, the vital viral proteins of FMDV that could affect the expression of HDAC5 in HEK293T cells were screened, and the expression of HDAC5 after co-transfection with viral proteins of FMDV and HDAC5 was detected. As shown in Figure 5A, the expression of HDAC5 was significantly reduced by VP1 and 3C. The decrease of HDAC5 induced by VP1 was further verified. As shown in Figure 5B, VP1 reduced the expression of HDAC5 in a dose-dependent manner. To investigate whether there was an interaction between VP1 and HDAC5, immunoprecipitation was performed using an anti-Myc antibody. As shown in Figure 5C, HDAC5-Myc precipitated VP1-Flag, and the expression of HDAC5 was decreased by VP1 compared with that in the control groups. Converse immunoprecipitation was performed using an anti-Flag antibody, and we found that VP1-Flag also interacted with HDAC5-Myc and reduced the expression of HDAC5 (Figure 5D). These results indicated an interaction between HDAC5 and VP1. Furthermore, the reduction in HDAC5 expression was reversed by the proteasome inhibitor, MG132 (Figure 5E). These results indicate that VP1 degrades HDAC5 via the proteasomal pathway.

## 4. Discussion

Studies have reported that class IIa HDACs, such as HDAC4, HDAC5, HDAC7, and HDAC9, are involved in the regulation of innate immune responses and viral replication [34,35,36]. Furthermore, HDAC4 has been shown to restrict VACA and HSV-1 replication by promoting IFN-I signaling and is targeted for proteasomal degradation by the VACA C6 protein in HEK293T and HeLa cells [28]. The expression of HDAC4 was found to enhance resistance to VSV through the activation of STAT1 in Hep3B core cells but was downregulated by HCV core protein [37]. HDAC9 was shown to enhance TBK1 kinase activity by interacting with and deacetylating TBK1 at the lys241 site, eventually promote the production of IFN-I [38]. Although HDAC5 has been reported to prevent the replication of DNA viruses such as VACV, HSV-1, and HBV [39,40,41], its role in the replication of RNA viruses and its mechanism have yet to be fully elucidated.

In this study, the deficiency of HDAC5 was found to markedly suppress IRF3/IFNβ-mediated signaling while promoting viral replication, mRNA expression, and the titers of FMDV and VSV. Meanwhile, the overexpression of HDAC5 in HEK293T cells upon poly(I:C) treatment was observed to promote the mRNA expression of IFNβ, ISG15, and ISG56, as well as those of IFN-α, IL6, IL8, and ISG54, indicating that HDAC5 may be a common target for many RNA viruses, reflecting its potential as a pan-antiviral. The replication of FMDV was inhibited by HDAC5, while the FMDV capsid protein VP1 counteracted HDAC5-meidiated activation of antiviral immune responses and facilitated viral replication by degrading HDAC5. A previous study reported that the deregulation of HDAC5 by IRF3 is crucial for the induction of lymphangiogenesis caused by KSHV. However, the study only reported the interaction between IRF3 and HDAC5 and did not investigate the specific regulatory mechanism of HDAC5 on IRF3/IFNβ-mediated signaling [39]. In our study, the deficiency of HDAC5 inhibited the phosphorylation of IRF3, and no interaction was observed between HDAC5 and IRF3 in HEK293T cells, which may be due to the different cell types or experimental treatments used.

To screen which FMDV viral proteins interact with HDAC5, coimmunoprecipitation was performed. Among FMDV viral proteins, the most obvious degradation of HDAC5 was mediated by 3C, but no interaction was observed between 3C and HDAC5. So VP1 was selected for further study. The FMDV protein VP1 suppresses the IFN-I signaling pathway by interacting with sorcin and counteracts the antiviral immune responses of TPL2 mediated by reducing the phosphorylation of TPL2 at Thr290 [11,12]. In the present study, VP1 was found to degrade HDAC5 in a dose-dependent manner. VP1 interacted with and degraded HDAC5. Moreover, rescue of HDCA5 protein degradation occurred upon VP1 overexpression with MG132 treatment, indicating that VP1 enhances the degradation of HDAC5 through the proteasome pathway. However, additional studies are needed to elucidate the specific mechanisms underlying HDAC5 degradation.

FMD, caused by the FMDV, is a serious infectious disease with devastating economic consequences [42]. Vaccination remains the most effective method for disease prevention and control [43,44]. Therefore, improving FMDV amplification and vaccine production is essential [45]. To improve FMDV amplification and vaccine production, it is crucial to maintain a stable viral copy number and expand the number of PK-15, BHK-21, and IBRS-2 cells. BHK-21 cells derived from one-day-old mice are used for the proliferation of viruses because of their fast growth rate and broad virus-sensitive spectrum, which is also recommended by the WTO for FMDV culture and the production of inactivated FMDV vaccines. CRISPR-Cas9 technology has been widely used not only to regulate gene expression, viral replication, and virus-induced immune responses, but also to promote the production of vaccine virus. In our study, the HDAC5-KO BHK-21 cell line was established using CRISPR-Cas9 technology. In contrast to the wild-type cells, HDAC5-KO BHK-21 cells exhibited no noticeable variance in their growth rate, but showed a higher viral titer, which implies potential for FMDV vaccine production.

In conclusion, this study demonstrated that HDAC5 activated the IRF3-mediated innate immune response against RNA viruses, such as FMDV. To counteract the antiviral effect of HDAC5, FMDV VP1 attenuated the HDAC5-mediated antiviral immune response by interacting with and degrading its expression through the proteasome pathway. These findings highlight the significance of the host protein HDAC5 in the regulation of RNA viral replication and provide novel insights into the mechanisms by which FMDV VP1 evades the antiviral immune response to facilitate viral replication, opening new avenues for increasing viral vaccine production and developing new antiviral therapies targeting this host protein.

## Figures and Tables

**Figure 1 cells-13-00539-f001:**
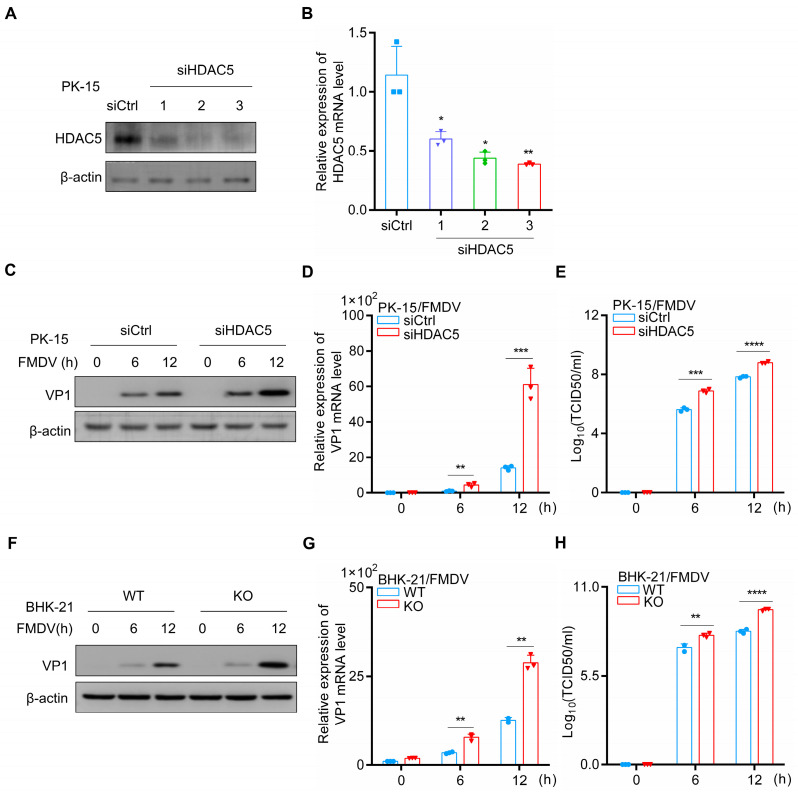
The deficiency of HDAC5 promotes FMDV proliferation. (**A**,**B**) siCtrl or three HDAC5 siRNAs were transfected into PK-15 cells (2 × 10^5^) and incubated for 48 h. Subsequently, the cells and culture supernatants were separately harvested at the indicated time points. (**A**) Western blotting was used to evaluate β-actin and HDAC5 protein expression. (**B**) RT-qPCR was applied to quantify the mRNA levels of HDAC5. (**C**–**E**) siCtrl or HDAC5 siRNA-3 (siHDAC5) was transfected into PK-15 cells (2 × 10^5^) and incubated for 36 h, followed by FMDV infection. Subsequently, the cells and culture supernatants were separately harvested at the indicated time points. (**C**) Western blotting was used to evaluate β-actin and VP1 protein expression. (**D**) RT-qPCR was applied to quantify the mRNA levels of VP1. (**E**) TCID_50_ was employed to test the viral titers of FMDV. (**F**–**H**) WT and HDAC5-KO BHK-21 cells (1 × 10^6^) were infected with FMDV. Subsequently, the cells and culture supernatants were separately harvested at the indicated time points. (**F**) Western blotting was used to evaluate β-actin and VP1 protein expression. (**G**) RT-qPCR was applied to quantify the mRNA levels of VP1. (**H**) TCID_50_ was employed to test the viral titers of FMDV. Groups were compared by unpaired Student’s *t*-test. *p* < 0.05 *, *p* < 0.01 **, *p* < 0.001 ***, *p* < 0.0001 ****.

**Figure 2 cells-13-00539-f002:**
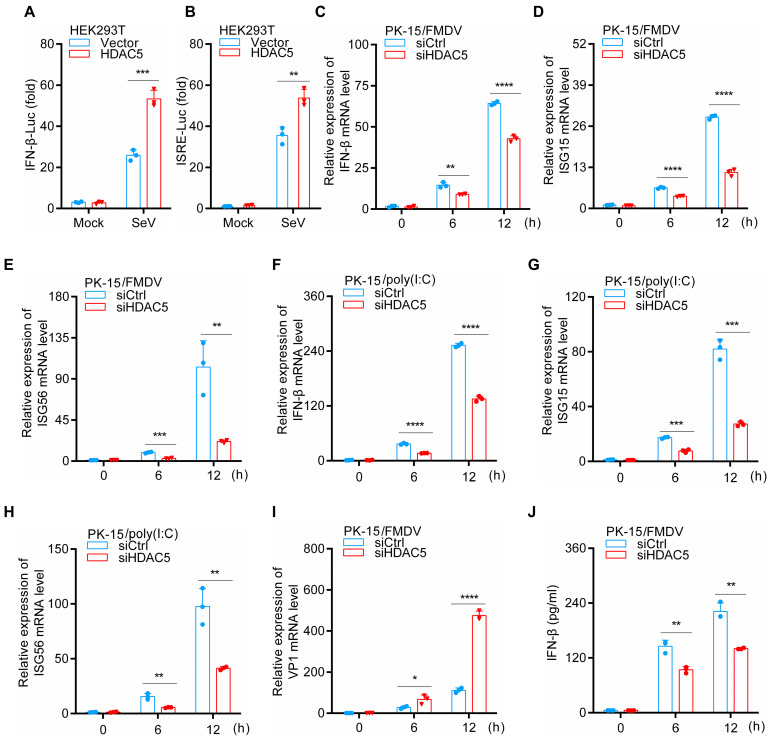
HDAC5 regulates the FMDV-triggered production of IFN-β and ISGs. (**A**,**B**) vector-control or HDAC5 plasmids were transfected into HEK293T cells (1 × 10^5^), as well as IFN-β or ISRE reporter plasmids, and incubated for 24 h. After SeV stimulation for 12 h, a dual-specific luciferase assay was used to detect activity of the IFN-β or ISRE reporter system. (**C**–**E**) siCtrl or siHDAC5 was transfected into PK-15 cells (2 × 10^5^), followed by treatment with FMDV. Subsequently, cells were separately harvested at the indicated time points. RT-qPCR was applied to quantify the mRNA levels of IFN-β, ISG15, and ISG56. (**F**–**H**) siCtrl or siHDAC5 was transfected into PK-15 cells (2 × 10^5^), followed by treatment with poly(I:C) (2 μg/mL). Subsequently, cells were separately harvested at the indicated time points. RT-qPCR was applied to quantify the mRNA levels of IFN-β, ISG15, and ISG56. (**I**) RT-qPCR was applied to quantify the mRNA levels of VP1 (as in (**C**–**E**)). (**J**) Culture supernatants were separately harvested at the indicated time points, and ELISA was employed to detect IFN-β secretion (as in (**C**–**E**)). Groups were compared by unpaired Student’s *t*-test. *p* < 0.05 *, *p* < 0.01 **, *p* < 0.001 ***, *p* < 0.0001 ****.

**Figure 3 cells-13-00539-f003:**
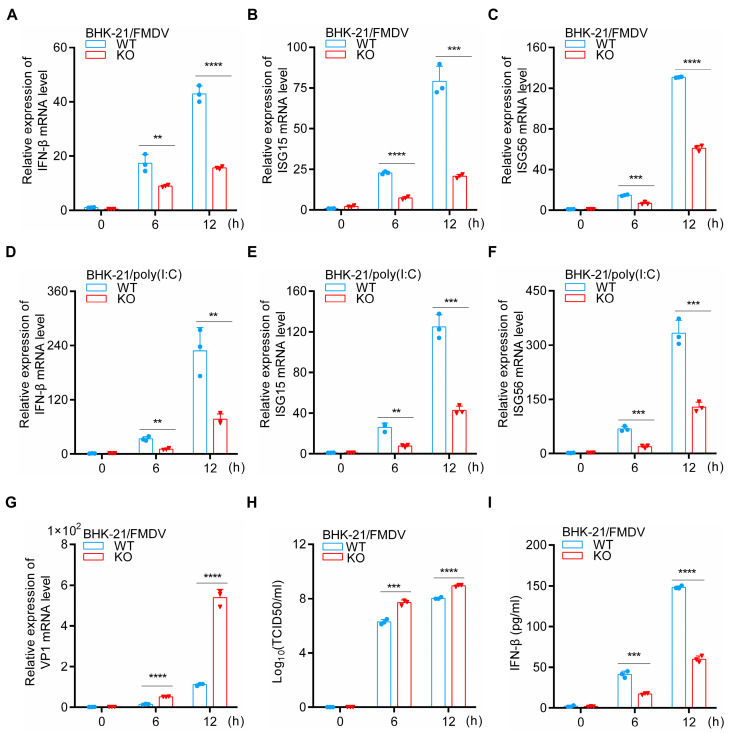
HDAC5 knockout inhibits the FMDV-, and poly(I:C)-stimulated production of IFN-β and ISGs. (**A**–**C**) WT and HDAC5-KO BHK-21 cells (1 × 10^6^) were infected with FMDV. Subsequently, cells were harvested separately at the indicated time points. RT-qPCR was used to quantify the mRNA levels of IFN-β, ISG15, and ISG56. (**D**–**F**) WT and HDAC5-KO BHK-21 cells (1 × 10^6^) were transfected with poly (I:C) (2 μg/mL). Subsequently, cells were harvested separately at the indicated time points. RT-qPCR was applied to quantify the mRNA levels of IFN-β, ISG15, and ISG56. (**G**) RT-qPCR was applied to quantify the mRNA levels of VP1 (as in (**A**–**C**)). (**H**) TCID_50_ was employed to test the viral titers of FMDV (as in (**A**–**C**)). (**I**) Culture supernatants were harvested at the indicated time points, and ELISA was employed to detect IFN-β secretion (as in (**A**–**C**)). Groups were compared by unpaired Student’s *t*-test. *p* < 0.01 **, *p* < 0.001 ***, *p* < 0.0001 ****.

**Figure 4 cells-13-00539-f004:**
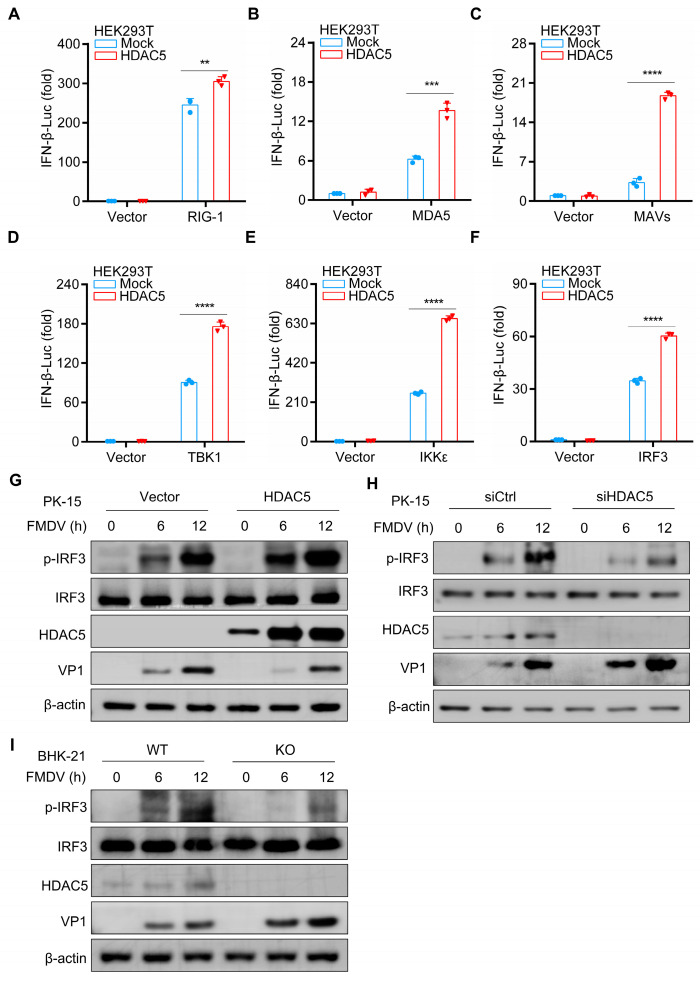
HDAC5 activates the FMDV-induced phosphorylation of IRF3. (**A**–**F**) Plasmids as indicated were transfected into HEK293T cells (1 × 10^5^) and incubated for 24 h. A dual-specific luciferase assay was used to detect the activity of the IFN-β reporter system. (**G**) HDAC5 or empty-vector plasmids (2 μg) were introduced into PK-15 cells (2 × 10^6^) and incubated for 36 h, followed by FMDV infection. Subsequently, the cells were harvested separately at the indicated time points at specific time points. Western blotting was used to evaluate the protein expression of phosphorylated IRF3, total IRF3, HDAC5, VP1, and β-actin. (**H**) siCtrl or siHDAC5 was transfected into PK-15 cells (2 × 10^6^) and incubated for 36 h, followed by FMDV infection. Subsequently, the cells were harvested separately at the indicated time points. Western blotting was used to evaluate the protein expression of phosphorylated IRF3, total IRF3, HDAC5, VP1, and β-actin. (**I**) WT and HDAC5-KO BHK-21 cells (1 × 10^6^) were stimulated with FMDV. Subsequently, the cells were harvested separately at the indicated time points. Western blotting was used to evaluate the protein expression of phosphorylated IRF3, total IRF3, HDAC5, VP1, and β-actin. Groups were compared by unpaired Student’s *t*-test. *p* < 0.01 **, *p* < 0.001 ***, *p* < 0.0001 ****.

**Figure 5 cells-13-00539-f005:**
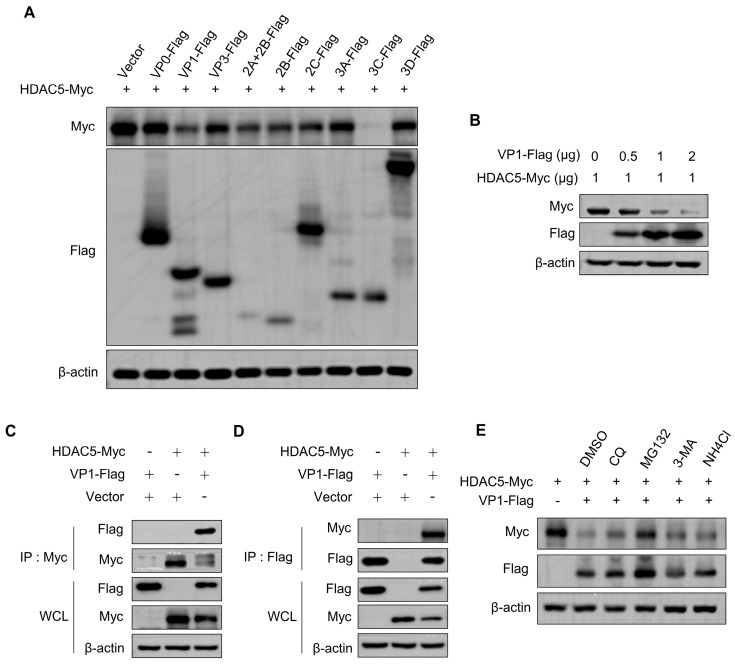
FMDV protein VP1 interacts with and degrades HDAC5 though the proteasome pathway. (**A**) Plasmids (1 μg) as indicated were transfected into HEK293T cells (1 × 10^6^) and incubated for 36 h. Subsequently, the cells were harvested separately. Western blotting was used to evaluate the protein expression of HDAC5, VP0, VP1, VP3, 2A + 2B, 2B, 2C, 3A, 3C, 3D, and β-actin. (**B**) Plasmids as indicated were transfected into HEK293T cells (6 × 10^6^) and incubated for 36 h. Western blotting was used to evaluate the protein expression of HDAC5, VP1, and β-actin. (**C**,**D**) HDAC5-Myc, VP1-Flag, and empty-vector (5 μg) were transfected into HEK293T cells (6 × 10^6^) and incubated for 36 h. Subsequently, the cells were harvested separately. Western blotting was performed on both whole-cell lysates (WCL) and IP complexes using specific antibodies. (**E**) HDAC5-Myc, VP1-Flag, and empty-vector (1 μg) were transfected into HEK293T cells (1 × 10^6^) and incubated for 36 h, followed by DMSO, MG132 (10 μM), CQ (100 μM), NH_4_Cl (5 M), and 3-MA (10 mM) treatment. Subsequently, the cells were harvested separately. Western blotting was used to evaluate protein expression of HDAC5, VP1, and β-actin.

**Table 1 cells-13-00539-t001:** RT-qPCR primers.

Gene Name	Sequence (5′-3′)
Human HDAC5	CCACGCTAGATGAGATCCAGACAG
	CACAAGGCAGCACAGCATACATC
Human IFN-β	GGACGCCGGACGCCGCATTGACCATCTATG
	ACAATAGTCTCATTCCAGCCAGTGC
Human GAPDH	GTGACGTTGACATCCGTAAAGA
	GCCGGACTCATCGTACTCC
Human ISG56	TCATCAGGTCAAGGATAGTC
	CCACACTGTATTTGGTGTCTAGG
Human ISG15	GGAATAACAAGGGCCGCAGCAG
	AGGTCAGCCAGAACAGGTCGTC
VSV-G	CAAGTCAAAATGCCCAAGAGTCACA
	TTTCCTTGCATTGTTCTACAGATGG
Mouse IFN-β	CAGCTCCAAGAAAGGACGAAC
	GGCAGTGTAACTCTTCTGCAT
Mouse ISG15	TCCTGGTGTCCGTGACTAACTC
	AAGACCGTCCTGGAGCACTG
Mouse ISG56	TGAGATGGACTGTGAGGAAGGC
	TCTTGGCGATAGGCTACGACTG
Mouse GAPDH	AGGTCGGTGTGAACGGATTTG
	TGTAGACCATGTAGTTGAGGTCA
FMDV-VP1	GACAACACCACCAACCCA
	CCTTCTGAGCCAGCACTT
Sus IFN-β	ACCTACAGGGCGGACTTCAA
	GTCTCATTCCACCCAGTGCT
Sus ISG15	ATGGGCTGGGACCTGACGG
	TTAGCTCCGCCCGCCAGGCT
Sus ISG56	ACGGCTGCCTAATTTACAGC
	AGTGGCTGATATCTGGGTGC
Sus β-actin	GCTGGCCGGGACCTGACAGACTACC
	TCTCCAGGGAGGAAGAGGATGCGGC

**Table 2 cells-13-00539-t002:** siRNA sequences targeting Sus HDAC5 gene.

siRNA Name	siRNA Sequence (5′-3′)
siHDAC5-1	GUCCAGUGCUGGUUACAAATT
	UUUGUAACCAGCACUGGACTT
siHDAC5-2	GGCAAGUUCAUGAGCACAUTT
	AUGUGCUCAUGAACUUGCCTT
siHDAC5-3	CCACGCUAGAGAAAGUCAUTT
	AUGACUUUCUCUAGCGUGGTT
siRNA control	UUCUCCGAACGUGUCACGUTT
	ACGUGACACGUUCGGAGAATT

## Data Availability

The study encompasses the original findings outlined in this article and the Appendix A. Additional inquiries can be directed to the corresponding author.

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
