# Peer review of "Foot-and-Mouth Disease Virus Capsid Protein VP1 Antagonizes Type I Interferon Signaling via Degradation of Histone Deacetylase 5"

_cells, 2024, doi:10.3390/cells13060539_

Round 1

Reviewer 1 Report

Comments and Suggestions for Authors

In this draft, the authors presented very interested results highlighting the ability of VP1 to influence IFN pathway. The presented results have also putative outcomes in vaccines’ production. The design of the experiments is really well done, and I found only few criticisms:

1-      Lines 35-36. STA instead SAT (South African Territories)

2-      The references of Introduction are not accurate. Eg: talking about FMDV NSP you cited a paper about Senecavirus (#ref 7) and talking about capsid you cited an article focused on single-cell analysis (#ref 9). An overall revision of References is mandatory.

3-      Line 42. Repetition VP1 VP1. Actually VP1-2-3 make up the entire outer surface and not most of it as you said.

4-      Lines 54-68. A lot of info without further explications and without a logical thread. All that information is difficult to fix in the memory and I am asking to myself if they are all very important to understand the whole paper. In my opinion this paragraph needs to be rephrased and probably a preventive selection of the info to be included should be done.

5-      Line 165. It is better to clarify here the line of KO cells. Moreover, all the paragraph (165-169) should be moved in line 171 after “We further assessed the propagation of FMDV in HDA5-KO BHK-21 cells.” So far you introduce the cell line and indicate clearly the aim of the experiment, that needs for KO. Then you will explain how you obtained the CRISPR-Cas9 engineered line.

6-      Why did you decide to compare FMDV and VSV. Two RNA viruses but one (+) and the other (-). You should explain this decision in the text.

7-      Comparing graphs 1D-1E and 1G-1H a huge difference between increase of VP1 mRNA and TCID50 is evident. I mean that KO and siRNA showed a real increase of VP1 mRNA while the difference in viral concentration between treated and control is not so huge. Can you discuss it?

8-      Line 228. Probably the position of the sentence “as well as the secretion of IFN-beta protein” is wrong. Reading that sentence, I understood that you noticed a promotion of virus copy and also an increase of IFN secretion. But looking at the graphs the IFN decrease both in expression and secretion.

9-      Line 235. It’s better to clarify that HEK293T cells are transfected with HDAC5 plasmid.

10-   In figure 4A-F you did not clarify if HEK293T were infected with FMD or not. If I am not wrong you said that in the text (line 257) but not in the picture.

11-   Figure 5A. I notice a strong reduction of 3C. Could you discuss it in the Discussion?

12-   In the Discussion you did not argue about the reduction of pIRF3. You could add this info in lines 331-332.

Reviewer 2 Report

Comments and Suggestions for Authors

Overall I find this a tidy study with some interesting observations. The VP1 protein of FMDV has previously been found to restrict the innate immune response in host cells. I do however have a few comments.

1) I am curious why VSV is occasionally discussed. There are no data within the main text; it is simply mentioned in passing throughout the manuscript. This seems somewhat confusing to me and in a way muddies the story.

2) In Fig 5A by far the most conspicuous decrease in HDAC5 (myc) is observed during in co-expression of 3C. The logical avenue to explore would therefore appear to be 3C. However, the authors proceed to investigate the impact of VP1 without even mentioning this result. I think the authors need to justify why they selected VP1 and not 3C.

3) The authors make a claim that HDAC5-KO cells may represent a useful cell line to generate vaccines as it would release some of the restriction on FMDV. The data provided for this claim is provided in Fig 3G. However, the data is simply RNA expression. I think it would be preferable here to quantitate the infectious virus output from these cells as compared to wild-type cells. this would allow a much clearer indication as to the usefulness of the cells for vaccine growth.

4)  With the exception of Figs 1 and 2 the majority of the work is undertaken in BHK (hamster) or HEK293T (human) cells rather than a cell line derived from a relevant species. I think that the authors should cover this point in the discussion.

5) I don't see details of the SeV experiments in the materials and methods.

Comments on the Quality of English Language

Overall I find the English to be good. There are occasions where the English is not quite correct, for example lines 156-158 and 259-261. 

Reviewer 3 Report

Comments and Suggestions for Authors

This is a nice demonstration of the role of HDAC5 in FMDV replication and the linkage of HDAC5 overexpression in IFN responses.

Line 42 replace the second VP1 with VP2

Line 356 remove and VSV.

Reviewer 4 Report

Comments and Suggestions for Authors

Gong et al present an interesting and mostly well-written manuscript exploring the role of histone deacetylase 5 (HDAC5) in interferon signaling in the context of foot and mouth disease virus infection.  The data support that VP-1 may inhibit interferon signaling by promoting HDAC5 degradation.

The major concern with the manuscript is that the materials and methods section is missing key details that would help explain the experiments and support the importance of the findings.

Specific concerns: how long prior to infection are cells treated with PolyI:c? Were the western blots stripped and re-probed or separate gels using the same lysates?

Other concerns include labeling of figure legends.  It is difficult upon reading to determine how some of the panels and treatments vary when the axes are identically labeled for many of the figures.  For example, Figure 3A-3C are different than 3D-3F, but are difficult to follow and thus reduce the impact of the figure, which is also true for Figure 4.  Suggest adding the treatment condition next to the cell type name in the upper left corner of all figures.  Also recommend using the same scale for each axis that is the same.

An experiment that must be considered is determining if VP1 can act as a ubiquitin ligase, with the understanding that such an experiment may be considered beyond the current scope of the work.  It would benefit the paper to include data supporting that in the presence of VP1 HDAC5 is ubiquitinated.

The discussion would benefit from expanding on on HDAC5 may be a common target for many viruses and its potential as a pan-antiviral/target for viral proteins to dampen interferon response.  Since it is a histone deacetylase, is it found more commonly at in ISG and IFN genes or other 'quick response' genes to be responsive to such stimuli?

Comments on the Quality of English Language

Overall, the English is fine, but an additional proofreading would help.  The figure legends state: 'subsequently, cells were harvested separately at specific time points' recommend changing throughout the manuscript and where applicable a shorter phrase such as "harvested at the indicated time points".

Instead of using the phrase "obviously" provide the fold change.  Also consider "markedly" instead of "obviously". 

Also suggest that the Materials and Methods and results be restructured so that many of the experimental details are removed from the results and figure legends and written into the Materials and Methods.  Based on how the Materials and Methods are written, I do no think I could repeat these experiments.

Round 2

Reviewer 1 Report

Comments and Suggestions for Authors

No other issues detected. Well done. Very interesting work.